# Aesthetic Criteria in Fundamental Physics—The Viewpoint of Plato

**Ivan Melo**

Department of Physics, FEIT, University of Žilina, Univerzitná 1, 010 26 Žilina, Slovakia; ivan.melo@feit.uniza.sk

**Abstract:** I discuss the role of beauty in physics. Physicists are sometimes described as platonists for their conviction that the fundamental laws are elegant and aesthetic arguments represent an important epistemic tool. After a review of the ideas of Plato and some of the leading figures of modern physics, which suggest that this is indeed the case, I present a list of current aesthetic criteria. I focus on symmetry and unity and demonstrate their increasing relevance in an array of experimentally verified theories. I also deal with an open issue of naturalness as the third criterion. The laws of nature appear increasingly beautiful as we uncover them, and it is important to see if the trend continues in the future.

**Keywords:** beauty in fundamental laws; aesthetic criteria in physics; symmetry; unity; naturalness

## 1. Introduction

Physics brings deep philosophical insights into the nature of the universe. One of them is the role of beauty in natural science. Throughout history, many leading physicists contemplated this role. Some claimed beauty was an inspiration for them to do science, another group argued beauty is a signpost to the truth and a few ventured to say they preferred beauty to truth in their work. Although they did not agree on all points, one thing seems clear: from Kepler to Einstein, Dirac, Weinberg, and Arkani-Hamed, aesthetic arguments have always been important. A recent study has found that 75% of respondents, mostly physicists and biologists, encounter beauty in their research [1]. According to the survey, beauty motivated most of the participants to pursue a scientific career, helps them persevere through difficult times, and most scientists believe beauty is key to communicating science. Given such a huge influence, the amount of literature written on the subject is relatively modest (for a review see [2,3]). The burden of rigorous treatments is on the shoulders of philosophers of science, e.g., [4–6], while physicists, if at all, prefer to express their views in popularization books [7–11] or talks [12–15].

Part of the reason is the perception of beauty as being in the eye of the beholder, a valid position, but the one which seems to make meaningful discussion difficult. What use is a vague concept in a rigorous discipline? On the other hand, when physicists who describe fundamental laws as profoundly beautiful are prompted to be specific, they give descriptions that frequently agree—a possible indicator of the existence of an objective element of beauty. Moreover, they do not just perceive aesthetic properties such as simplicity, symmetry, and unity in the laws of physics, but also actively use them as key ingredients to building new theories, elevating beauty well beyond a decorative role. Two discoveries in the last decade, both leading to the Nobel prize, stand as milestones confirming this practice. The Higgs boson, discovered by the ATLAS and CMS collaborations at the Large Hadron Collider (LHC) at CERN in 2012, was the triumph of the Standard Model of particle physics, and gravitational waves, detected by the Advanced LIGO detectors in 2015, paid tribute to Einstein's General relativity.

The treatment of beauty as an objective entity has its roots in the works of Plato. He associated beauty with knowledge, in particular the knowledge of mathematics and philosophy. Beauty was, along with goodness, central to Plato's philosophy since it provided a

connection between the physical world and the world of ideas (forms) where true knowledge resides. His claim that the object of education is to teach us to love what is beautiful echoes the finding of the survey about beauty's key role in communicating science. As a further sign of his lasting influence, two of the five conceptions of beauty listed by the Stanford Encyclopedia, idealist and classical, can be linked to Plato directly or through his student Aristotle. Both are particularly useful in physics. Plato was also an accomplished mathematical physicist who believed in the mathematical structure of the Universe and developed an authoritative theory of the building blocks of matter in terms of Platonic solids. I agree with G.C. Field who wrote that the physicist familiar with Plato's philosophy finds it very friendly to his own habit of thought [16]. We will see many indeed do.

I will reflect here on the role of beauty in fundamental physics, using the general framework of Plato as background. By fundamental physics I mean the physics of the four fundamental interactions: electromagnetic, strong, weak, and gravitational, i.e., particle physics plus gravity. I support the view that, as far as we know, the universe is an elegant and orderly place at the fundamental level. Frank Wilczek made an important note: this orderly elegance is far from obvious [17]. It could have been otherwise. It is possible to imagine a chaotic world with one law applying here, another one there, no connections among different phenomena and rules changing all the time. It is also far from trivial. The fundamental laws rest on simple principles, yet they are deep, and many rich and complex structures derive from them. Just as art, physics lifts us and shows everyday life in the light of ideal—everyday phenomena in the light of beautiful theories. This message is worth sharing with many audiences, from students to researchers, and, potentially, to the interested members of the general public. My treatment, written from the perspective of a particle physicist, is for researchers in various disciplines. Given that beauty is a powerful source of motivation for us scientists, we should strive to grasp its meaning.

I will start with a description of Plato's ideas in Section 2. Then, in Section 3, we will discuss opinions held by both the founders of modern physics and their successors. In Section 4, I describe aesthetic criteria that physicists use (along with empirical criteria) both in theory development and theory evaluation. I emphasize unity and symmetry, complemented by naturalness. Subsequently, in Sections 5 and 6, I show how they apply to individual physical theories. I focus on established and empirically confirmed theories up to the Standard Model of particle physics, a firm basis with an excellent aesthetic record. The discussion will bring us to the naturalness problem of the Standard Model, an open issue that could have either beneficial or detrimental outcomes for the status of beauty in science. The role of beauty in McAllister's theory of scientific revolutions is addressed in Section 7. I conclude with a summary of the main points.

## 2. Plato's Theory

The basic problem in the theory of beauty is whether beauty is objective or subjective. The former, objectivist position, according to which beauty is located in the beautiful object itself, outside of any observer, has its origins in the teaching of Plato. It dominated the Western world until the 18th century. An alternative view, the subjectivist position, was described by David Hume in the following way: "Beauty exists merely in the mind which contemplates them, and each mind perceives a different beauty" [18]. Likewise, Immanuel Kant [19], challenging the founder of aesthetics Baumgarten [20], wrote that judgment of taste and beauty is not a judgment of cognition, and, consequently, is not logical but subjective. A pure version of either of these positions seems implausible [21]. There are arguments for and against each side with no clear resolution.

If we turn to physics, many physicists, consciously or not, identify with the objectivist account. Jesus Zamora Bonilla calls them platonists and argues that for them, beauty is an essential ingredient in the process of scientific research. On the other hand, skeptics deny that scientific research has anything intrinsic to do with beauty. The first group is, according to Zamora Bonilla, dominated by quantum physicists and mathematicians, the second one by other scientists (and philosophers) [22].

Skeptics among physicists often adhere to a conviction that observation is our only reliable source of knowledge (empiricism). Our theoretical models should, in the first place, describe experimental data and predict new phenomena (instrumentalism). Their explanatory role is appreciated but plays just second chair [23]. Sceptics would use quantum mechanics for illustration because of its usefulness, despite the lack of agreed-on interpretation.

In contrast to that, platonists would likely argue that both observation and reason are indispensable sources of knowledge, and that, in a sense elucidated below, pure reasoning can grasp reality. Explanation and thorough understanding offered by a theory are both at least as important as the agreement with facts. Despite Kant, the judgment of beauty is a judgment of cognition. And, some would say, beauty guides reason to knowledge and truth.

Let us take a closer look at Plato's teaching with a motive to get a better understanding of one view on the role of beauty in physics. He makes a distinction between the sensible (material, physical) world and the world of forms which represents a more ultimate reality that can be known only by reason [24]. The forms are ideas or ideals which express the essence of many individual things. For example, a mathematical expression for the circle is a form, while many circles drawn on a piece of paper are just imperfect images of this single form. Hence the material objects share some features of their ideal forms but not all of them. The sensible world is understood by observation and reason, the forms by pure reasoning only. The former is the world of change and decay, the latter of unchanging reality.

Plato had a preference for the world of forms, the knowledge of which he thought superior to the knowledge of the sensible world. True wisdom is the knowledge of the forms. He used the allegory of the cave to explain this further. The prisoners, held in the cave without any movement from birth, watch the passing shadows of the outside world on the wall, unable to realize that the shadows correspond to real animals and objects. Their sensible world is the world of shadows. The knowledge of the real world outside could be inferred from the shadows by pure thought if the prisoners had a philosophical spirit.

The reason Plato valued beauty so much was that he saw it as an entrance to his ultimate reality of forms, a way to get out of the cave. Beauty, unlike other forms, has the privilege of being clearly visible in the sensible world [25]. As a result, it has an excellent potential to guide every mind from the sensual experience to the knowledge of the higher forms [21], the real knowledge. In the Republic, Plato asks if the knowledge of all other things is of any value if we lack knowledge of beauty and goodness. The object of education is to teach us to love what is beautiful, from individual things to beauty itself [24].

The forms exhibit hierarchy. The highest form of beauty is perfect, eternal, and unchanging, beauty in itself. Then the lower forms of beauty follow, being the less and less faithful copies of the ideal form. At the bottom level, we find the things and phenomena of the sensible world. Beautiful as they may be, they are just imperfect images of the higher forms of beauty.

In Symposium Plato gives his definition of beauty as perfect unity, universal beauty sitting at the top of a ladder. It can be reached rung by rung starting with the contemplation of many individual beauties, moving from the beauty of the outward form to the beauty of the mind, i.e., from the beauty of the human body to the beauty of laws and institutions, and on to the sciences, and finally to the top where one will see tremendous beauty [26]. This is an important point. Plato believes that absolute beauty, perfect unity, exists and can be known.

The laws of physics as we understand them today are close to the definition of the forms. They have their independent existence and describe, in their unifying role, the essence of many seemingly unrelated phenomena. They are also unchanging, their mathematical formulation reflects Plato's belief in the mathematical structure of the universe, and, last but not least, they manifest hierarchy.

Associating beauty primarily with knowledge, rather than with emotions, Plato sets beauty on a (relatively) firm ground. It can be checked against empirical evidence and thus be a fruitful concept.

### 3. Platonists among Physicists

We can detect platonic views of beauty, to various degrees, in the thoughts of the leading figures of modern physics. We start with classic quotes from the founders of the field.

For Henri Poincare beauty was the motivation to do science and a guide to understanding: "The scientist does not study nature because it is useful; he studies it because he delights in it, and he delights in it because it is beautiful" [27]. He clarified he did not mean the beauty which strikes the senses, the beauty of qualities and appearances: "not that I undervalue such beauty, far from it, but it has nothing to do with science; I mean that profound beauty which comes from the harmonious order of the parts and which a pure intelligence can grasp" [27].

Werner Heisenberg thought of beauty as a pointer to the truth. He perceived it in the mathematical structures of great simplicity and admitted that it was hard to avoid thinking that such structures correspond to a real feature of nature [28]. Even usually pragmatic Feynman agrees that one can recognize truth by its beauty and simplicity, typically manifested in the observation that more comes out from a true theory than goes in [29]. Murray Gell-Mann spoke the same language: "What is especially striking and remarkable is that in fundamental physics a beautiful or elegant theory is more likely to be right than a theory that is inelegant. In 1957 some of us put forward a partially complete theory of the weak force, in disagreement with the results of seven experiments. It was beautiful and so we dared to publish it, believing that all those experiments must be wrong. In fact, they were all wrong" [14].

Hermann Weyl and Paul Dirac go even further, suggesting beauty has often a higher priority for them than truth. Weyl said: "In my work, I have always tried to unite the true with the beautiful; but when I had to choose one or the other, I usually chose the beautiful" [13]. He was pointing to his gauge theory of gravitation and electromagnetism which turned out to be wrong. His instinct, however, proved to be at least partially correct, for the gauge symmetry proved powerful in the context of the quantum theories of electromagnetic, strong, and weak interactions, as we will see later. Likewise, Paul Dirac wrote in 1963, commenting on the birth of Schrodinger's wave equation, the first version of which did not agree with experimental data, that beauty in equations is more important than their agreement with experiment [30]. In order to interpret this bold claim correctly, we add that he thought the discrepancy with the data could be caused by minor features in the theory that would get fixed later (as was the case with the Schrodinger equation).

Among the present-day physicists, Frank Wilczek says that we found beauty at the heart of the world, and beauty itself will guide us further [8]. Steven Weinberg thinks that theories of exceptional elegance and mathematical beauty are rarely entirely wrong. At worst, they turn out to be correct in a different context than they were invented for. He also made an interesting analogy in which he connects aesthetic properties to knowledge. When the horse breeder finds a horse beautiful, he is not, in Weinberg's opinion, expressing merely aesthetic emotions but the breeder knows from experience that this kind of horse wins races [7,31]. Just like Plato, Weinberg seems to distinguish an outer appearance, the lowest form of beauty, appreciated by every layperson, from the higher form, the true beauty of an object associated with a deep knowledge of it. We could continue with many more examples but let us conclude with Nima Arkani-Hamed who sees the laws of physics as crystalline, simple, deep things, which are almost perfect because it is hard to modify them. They appear inevitable [15]. And elsewhere he adds that inevitability is for him a substitute for beauty [11].

Besides motivation, beauty is described in three different roles, two epistemic and one heuristic. First, Heisenberg, Gell-Mann, Dirac et. al. link beauty to truth, giving it a prominent epistemic authority. It is important to note the nature of this link: while the truth is accompanied by beauty, not every beautiful theory leads to truth or better knowledge. McAllister, working with the notion of beauty as a subjective value, described a possible

dynamical mechanism behind the link in his theory of scientific revolutions [4]. I discuss this theory and add a platonist view in Section 7.

In a more recent development [3,32,33], it has been argued that the epistemic role of beauty might have a different character. Rather than truth, aesthetic properties could be linked to understanding. Crucial for this approach is the difference between truth and knowledge on one side and understanding on the other side, which appears to be the position of Poincare [32,34].

Finally, if not epistemic, beauty could take on a more modest, heuristic role as a guide in the choice between two empirically equivalent theories. For example, a simpler theory is easier to use than a complex one and this is indeed part of scientific practice.

A prominent epistemic authority naturally brings criticism from skeptics who either downplay it or doubt that epistemic judgments are genuinely aesthetic [35–39]. For example, Kosso argues that aesthetic value is entirely gratuitous to the epistemic role: "It is not that physicists find gauge symmetry to be beautiful, and on the basis of that beauty they believe it to be a real feature of nature. It is rather that, for mundane empirical reasons, physicists believe this symmetry is an aspect of nature, and, by the way, they also find it to be a thing of beauty" [35]. Borrelli writes that the heuristic successes in early particle physics were ascribed exclusively to symmetries only a posteriori, forgetting the role of other heuristic strategies [36], Todd suggests the possibility that aesthetic judgments are in fact masked epistemic judgements [37], and Kosso identifies beauty with intelligibility [38].

Other attempts which do not take aesthetic judgments as genuine and reduce them to empirical ones, were discussed by McAllister [4]. He opposed them and suggested that aesthetic criteria used by scientists should be considered truly aesthetic. I agree with him on this point. In this context an interesting result from neuropsychology, pointed out by Ivanova [3], demonstrates that mathematical beauty stimulates the same brain activity as music and art.

Einstein himself had doubts in his early years that we could discover the true laws of nature insisting on their mathematical simplicity and beauty. However, as pointed out by J.D. Norton, he changed his mind and converted to platonism after his painstaking experience with General relativity. In 1933, he said: "Experience may suggest the appropriate mathematical concepts, but they most certainly cannot be deduced from it. Experience remains, of course, the sole criterion of the physical utility of mathematical construction. But the creative principle resides in mathematics. In a certain sense, therefore, I hold it true, that pure thought can grasp reality, as ancients dreamed" [39].

Arkani-Hamed brings more light to this issue by distinguishing between laws and their formulations [15]. He represents different laws by local peaks in the theory landscape, which are perfect and hard to modify, one peak for classical physics, one for quantum physics, and so on. Since the peaks are well separated and not continuously connected, physicists have to make leaps to get from one to the next to make discoveries. Importantly, there are various, sometimes radically different, formulations of the same locally perfect law—different routes to the top of the peak. While all describe the same law, some formulations, so-called morally correct explanations, have an added value. Besides shining light on their own peak, they offer an excellent springboard for a leap to the next peak. This is the kind of explanation that physicists find the most beautiful. Arkani-Hamed gives, among others, the least action principle as an example. The original formulation of Newton's laws, deterministic in its nature, does not connect naturally to the probabilistic nature of quantum mechanics. But the least action formulation does and hence qualifies as morally correct. Morally correct explanations are very close to the deep and simple principles behind the fundamental laws. As a result, they are not just as locally perfect as the laws they explain, they are also likely to be correct for new theories [15]. They have the unifying power.

The approach of Arkani-Hamed offers a possibility to partially understand both platonic and skeptical positions: when facing a new physics peak, physicists can make a maiden ascent either via more 'mundane empirical' routes, finding bits and pieces of beauty along the way, or, like Einstein, via the morally correct route. The latter one is the

path of beauty pointing to the truth from start to finish. It should go without saying that both types of routes are legitimate.

## 4. Aesthetic Criteria

As we have seen, beauty is an important concept for many physicists as a motivational, epistemic, or heuristic tool. But what is that beauty they perceive like? What are its forms? Can one recognize it uniquely? Can it lead us astray?

The classical conception describes beauty as the proper conformity of the parts to one another, and to the whole [12,21]. The parts are arranged following certain order, proportion, harmony, or symmetry. Aristotle, one of the most influential figures behind this conception, wrote in Metaphysics: "The chief forms of beauty are order and symmetry and definiteness, which the mathematical sciences demonstrate in a special degree" [40].

The idealist conception centers around Plato's views described in Section 2. Beauty is perfect unity which, as forms do, exists outside the physical world and is independent of mental processes. Its traces can be found in the sensible world since physical objects and phenomena share some features of their ideal forms.

Both conceptions naturally accommodate mathematics. Heisenberg conjectured that they are not too far from each other [12]. Indeed, we might interpret the whole in the classical conception as platonic unity. This unity is, in physics, represented by a few principles expressed in mathematical forms which define the arrangement of parts, i.e., of many physical objects, processes, and phenomena. Other conceptions, which connect beauty with love, pleasure, use, and uselessness [21], seem to be of less relevance in physics.

Modern physicists suggested a variety of aesthetic criteria for theory evaluation. Steven Weinberg mentions four: simplicity, inevitability, symmetry, and rigidity [7]. Astronomer Thrinh Xuan Thuan lists three: conformity with the whole, simplicity, and inevitability [10]. Subrahmanyan Chandrasekhar puts forward two criteria: strangeness (meaning "exceptional to a degree that excites wonderment and surprise") and conformity of the parts to one another and to the whole [13]. Franck Wilczek proposed a different pair: productivity and symmetry [17]. Henri Poincare argued for simplicity and unity [34] and, finally, Anthony Zee suggests a single criterion, symmetry [9]. Thuan's and Zee's propositions were discussed at length, e.g., in [41]. Other criteria, elegance, sublimity, harmony, and wonder, are mentioned in Ref. [2].

Among these suggestions, I choose unity (or productivity) and symmetry for further discussion, supplemented by naturalness. Here are the reasons. Hunkoog Jho understands simplicity, symmetry, and unity as intrinsic aspects of aesthetics, whereas elegance, sublimity, and wonder are extrinsic, evoked from the intrinsic ones [2]. I agree with this position. Intrinsic criteria, with the exception of simplicity, are easier to define and treat mathematically, facilitating discussion. Symmetry and unity echo the two early Greek conceptions of beauty and I find them the most influential in theory building nowadays. Naturalness is not without controversies (Section 6) and I include its discussion because it illustrates how powerful beauty arguments in particle physics are, and how important it is to continue to investigate this question both in physics and the philosophy of science.

Wilczek calls a theory or a law productive, if, following its formulation from a set of clues and a guess, it can explain many other things, so in effect and in agreement with Feynman, more comes out than went in [17]. Typically, this takes the form of a few equations which unify and correctly describe a long list of phenomena. But there is more to it than that. What is crucial is the number and depth of principles, not the number of equations nor the number of particles. If a single principle inevitably and uniquely generates several equations which describe many particles and a multitude of phenomena, then productivity primarily derives from the principle and implies simplicity, inevitability, and, in particular, unity: the wide range of phenomena does not require a wide range of theories. Francis Hutcheson sees beauty in the "knowledge of some great principles, . . . from which innumerable effects do flow". [42]. Below we use productivity and unity as synonyms.

The second criterion, symmetry, has a role in the fundamental laws of physics which is hard to overemphasize. A theory has symmetry if its mathematical formulation does not change (remains invariant) under symmetry transformations such as rotations, translations, or reflections. Since the transformations change the objects the theory is constructed of, it is nontrivial that they do not change the theory itself in the process. Symmetry is rigorously defined in Group theory. As a mathematical concept, it is as objective as one could hope for.

Finally, the third influential aesthetic criterion in particle physics is naturalness. Naturalness is the belief that fundamental dimensionless parameters should be naturally of the order 1, or, more broadly, that fundamental parameters should not take random values but rather have a natural explanation. For example, Dirac was rightly bothered by the fact that the force of gravity between two protons is much weaker than the electric force, the ratio of the two being of the order $10^{-36}$ instead of 1. Likewise, particle physicists nowadays worry about the smallness of the Higgs mass with respect to the Planck mass, $M_H/M_P \sim 10^{-17} \ll 1$.

What makes naturalness powerful is its ability to imply that new physics is just around the corner. While unity and symmetry arguments may lead to the unification of electromagnetic, strong, and weak interactions at energies $\sim 10^{16}$ GeV, far from a direct experimental reach, the (un)naturalness of the Higgs mass suggests that new particles should be within the domain of the LHC at energies $\sim 10^3$ GeV. Being an influential criterion, the naturalness principle can become a legitimate target for criticism as we will see later.

Let us see how Plato applied the first two criteria to his geometric theory of the building blocks of matter. In Timaeus, he identified the four classical elements, fire, air, water, and earth with the four Platonic solids, the pyramid, octahedron, icosahedron, and cube, respectively. Platonic solids, representing atoms, are very symmetrical objects. Their faces, regular polygons (equilateral triangle, square, regular pentagon), all identical in shape and size, meet in an identical way at each vertex. While there is an infinite number of regular polygons, only five Platonic solids can be constructed out of them. The fifth solid, the dodecahedron, is, according to Plato, the shape of the Universe. Plato further goes into dynamical details such as the decomposition of water into fire and air: the icosahedron of water (with 20 sides) breaks down into two octahedra of air (8 sides each) and one pyramid of fire (4 sides) [43].

Wilczek praises the deep insight of Plato who saw great potential in the fact that symmetry leads to a small number of structures, anticipating thus the spirit of modern physics [44]. Further, as Zamora Bonilla pointed out [45], Plato in his theory unified the theory of the four elements with Democritus's theory of the atoms, and the Pythagorean idea about the decisive role of numbers (or mathematical structures) in nature. Despite Plato's theory being wrong, the two modern criteria, symmetry, and unity, shine here bright. As a bonus, Plato suggests that space ('receptacle') in which atoms move, is invisible stuff that is in constant chaotic motion, randomly interacting with the atoms [45]—in a striking resemblance to quantum vacuum. According to Karl Popper, this geometrical theory of the world has influenced our world picture to the point that we take it for granted [43].

Despite this achievement, Plato is sometimes viewed as an armchair philosopher, who downplayed the importance of observation, believing that all problems can be solved by pure thought alone. This portrayal may be an unfortunate result of his preference for the world of ideas over the world of senses. But he did not despise the latter. As demonstrated by others [16,43], Plato was well aware that observation and precise measurements were crucial both for applications and for providing material for the realm of ideas.

In the next section, we will give an overview of theories that exhibit increasing unity and symmetry. We will discuss the naturalness problem in the Standard Model of particle physics in Section 6.

## 5. Symmetry and Unity

One of the simplest symmetries is circular symmetry. The ancient Greeks believed that the planetary orbits must be exactly circular on the grounds of this simplicity. Given the

immense influence of Greek astronomy, the circular orbits enjoyed an undisputable status for almost two millennia. This picture was so deeply rooted in people's minds that they were reluctant to accept Kepler's conclusion that the planets move in ellipses.

Kepler himself, in his Platonic solids model of the Solar System, assumed circular orbits. He found that when the five Platonic solids were nested within one another, each encased in a sphere, one would get six spheres, corresponding to the six known planets of his time. The model yielded the ratios of the radii of the planetary orbits in rough agreement with observations. Kepler, struck by the beauty of his conception, set out to prove it by meticulously improving the precision of his calculations. In the end, the model was completely wrong. The distances of the planets are not fundamental parameters but rather historical accidents, and as such, they do not require a natural explanation. Naturalness did not lead Kepler to the result he had wished. His abandonment of the circular orbits in favor of elliptical ones is now considered to be a classic case of a scientific revolution [4] and an example of beauty misguiding science.

It was, however, beauty that fuelled his determination on the arduous path to the correct laws of planetary motion. Further, the Greek idea of a special role of circular symmetry was not completely wrong. Newton's law of gravitation is spherically symmetric, which includes circular symmetry. We learn an important lesson here. The real beauty is to be found not in the solutions (orbits in this case) of some law, but in the law itself. Solutions need not share the symmetries of the law. Since we typically see the solutions first, the full beauty remains hidden until the law is discovered. The unifying power of Newton's theory is manifested in that the single law of gravity describes both planetary motions and falling apples. Kepler's law of elliptical orbits and two other Kepler's laws are natural consequences of Newton's law.

Moving on to classical electromagnetism, the four Maxwell's equations exhibit several symmetries. In particular, their Lorentz symmetry takes us beyond classical electromagnetic phenomena. This symmetry ensures the laws of physics remain the same in any inertial frame. Once its importance was recognized by Einstein, it played a crucial role in the development of the Special theory of relativity, which unified space and time. The Lorentz symmetry became one of the key ingredients of future theories in fundamental physics and a powerful tool to significantly reduce the number of possible forms these theories could take. When Paul Dirac in the late 1920s insisted that quantum mechanics should be Lorentz invariant, he was ultimately led to the prediction of antimatter in the form of the positron. As for productivity, Maxwell's equations unify classical electricity, magnetism, and optics.

Respect for symmetry has further grown with the formulation of General relativity (GR). Einstein demanded that the laws of physics remain the same even in non-inertial frames, whatever their acceleration or rotation. The corresponding symmetry is known as the general covariance. Once Einstein understood the general covariance, there was not much freedom left, he was inevitably led to the correct answer. Symmetry dictated the structure of the theory and the existence of gravitation itself. The aesthetic property was no longer just an interesting or useful feature of equations, it became the principle from which the equations of the theory were derived.

The scope of phenomena united within the framework of GR is worthy of respect. Inspired by a single experimental fact (that different masses fall at the same rate) and based on the power of the general covariance, the theory includes Newtonian gravity in the non-relativistic limit and predicts, among others, the anomalous precession of Mercury's perihelion, the gravitational bending of light, gravitational time dilation, the gravitational redshift of light, black holes, expanding universe, and, gravitational waves. Further, while studying GR, Emma Noether discovered the powerful unifying link between symmetries and conservation laws: they are not separate entities but each continuous symmetry corresponds to a conserved physical quantity. Noether's insight meant that physicists could construct their theories with symmetries corresponding to observed conserved quantities, making their task much easier [9].

The circular symmetry, Lorentz symmetry, and the general covariance are all spacetime symmetries. Internal symmetries, being a different class, do not operate in spacetime but

in abstract inner space. Internal symmetries reached perfection in quantum field theories, such as the quantum theory of electromagnetism (QED). In QED, the inner symmetry responsible for electric charge conservation, the so-called $U(1)$ gauge symmetry (in essence, our familiar circular symmetry), is made local rather than global. As a consequence, gauge symmetry, like the general covariance, becomes the principle from which the equations of the theory are derived. The existence of the electromagnetic field itself follows from this principle. QED equations contain classical Maxwell's equations as a special case and a plethora of quantum effects including the existence of antiparticle (positron) and the annihilation of an electron and positron pair.

The local gauge symmetries play the same central defining role also in the Standard Model of particle physics, the state-of-the-art quantum theory of electromagnetic, weak, and strong interactions. Each of these three fundamental interactions is the consequence of a corresponding gauge symmetry. Moreover, electromagnetic and weak interactions are unified in the Standard Model into a single framework of electroweak interactions (albeit with two different strengths).

The trend towards unity in particle physics is clear. Besides what we have discussed, quantum theories also unify particles and fields in the sense that particles are quanta or excitations of the fields. There are ongoing attempts to go beyond the Standard Model and further unify particles by putting them in the same multiplet (like Heisenberg did with proton and neutron through isospin symmetry in 1932), to unify forces (Grand unification of electromagnetic, weak, and strong interactions) and to unify matter and forces (fermions and bosons via supersymmetry). It is not obvious that the trend will continue but it looks like Plato would like what we have achieved so far.

There are two remaining points I would like to address before moving on to naturalness. First, how do physicists reconcile obvious richness, diversity, and associated imperfections and asymmetry in objects and phenomena around us with the perfect symmetries of equations which describe them? There are 17 fundamental particles in the Standard Model and 19 free parameters. Most of the free parameters are the nonzero masses of the particles which pose a problem since they break otherwise perfect gauge symmetries. The solution was provided by the Higgs mechanism of symmetry breaking. According to the mechanism, the perfect symmetry was present both in the law and its solutions at very high temperatures in the early universe and the particles were massless. Then, as the temperature dropped below the critical temperature, the symmetry broke spontaneously and the particles became massive. The adjective 'spontaneous' means that the symmetry of the law/theory is always there, both above and below the critical temperature; what was broken was the symmetry of the solutions of the theory, in particular the symmetry of the solution with the lowest energy, the vacuum. The mechanism was confirmed by the discovery of a new particle, the Higgs boson, in 2012.

The spontaneous symmetry breaking could be a reason why the full beauty of nature remains veiled: it is much easier to see diverse, 'not-so-symmetric' solutions than the elegant theoretical framework behind them. We have touched this point before in the case of elliptic orbits in Newton's theory, here, however, we reached its mature form. Nature wants both unity and diversity, symmetry and the lack of it [9] and spontaneous symmetry breaking is a tool to connect the two.

The second point is a frequent claim by physicists that the Standard Model is 'ugly'. The adjective itself suggests the existence of the beautiful framework in physics, the standard with respect to which such aesthetic judgments can be made. It has been my intention here to show that the Standard Model itself is a crucial part of this framework. Physicists got accustomed to a high standard indeed, so they easily focus their attention on the remaining 'pimples'. As Frank Wilczek put it in his book, after we discovered beauty at the heart of the world, we hunger for more [8]. We long for the top of the ladder.

## 6. Naturalness

Naturalness is not a new principle. It was invoked early in the seventeenth century by Tycho de Brahe against the heliocentric model. The absence of the observable change of the

stars' relative positions as viewed from Earth as it orbits the Sun over the year (the parallax) led him to conclude that either the ratio of the Earth-Sun distance to the stars-Sun distance is very small or Sun orbits Earth. Tycho de Brahe found the former option unnatural [46] and arrived at the wrong conclusion.

Naturalness is, however, a valid aesthetic principle behind the problem of the smallness of the electron and proton masses. In classical electrodynamics, the electron's own electric field diverges at small distances and so does the energy in the field, the so-called self-energy. The self-energy contributes to the total electron's mass along with the bare mass of the electron (the mass which is not due to the electric field). Even if we make the electron non-pointlike fixing its radius to its upper experimental bound, the self-energy is still a million times greater than the measured electron's mass [47], which requires the bare mass to be fine-tuned to the self-energy: the two independent contributions must be unnaturally close to each other, to one part in a million, to cancel out and yield the small electron mass. Naturalness asks for a mechanism to understand this mysterious conspiracy. The explanation came in the form of chiral symmetry which forces the QED self-energy contribution to a natural value near the observed electron mass.

Our second example is the weakness of gravity when compared with electromagnetism. The ratio of the force of gravity between two protons to the electric force is unusually small, $F_g/F_e \sim 10^{-36}$. The problem can be reformulated as the smallness of the proton mass, $m_p \sim 1$ GeV/$c^2$, with respect to the Planck mass, $M_p \sim 10^{19}$ GeV/$c^2$, since $F_g/F_e = m_p^2/(M_p^2 \alpha)$ where $\alpha = 1/137$ is the fine structure constant. Dirac and Feynman thought there must be a reason behind and there was. Today we know the answer is provided by the proton substructure: the interactions of quarks and gluons grow very slowly (the property known as asymptotic freedom), i.e., naturally, on the long way down from high energies corresponding to $M_p$ to low energies where they become strong enough to form a bound state, the proton, at $m_p$.

Naturalness also explained the pion mass splitting and successfully predicted the charm quark mass [47]. It is perhaps not surprising that it was invoked again when physicists faced the problem of the smallness of the Higgs mass, $M_H = 125$ GeV/$c^2$, with respect to $M_p$. Here, at the frontiers of knowledge, the situation is less clear. The quantum contributions of the Standard Model particles drive the Higgs self-energy contribution to the Higgs mass all the way to the Planck mass and one is forced to demand an unnatural fine-tuning between the self-energy contribution and the bare Higgs mass to keep the theory in line with the measured value [48]. This problem, known as the naturalness problem in particle physics, motivated the search for new physics to an unprecedented degree.

In an effort to find a natural mechanism to explain the Higgs mass, scientists applied reasoning inspired by the electron and proton masses. One strategy was to use a new symmetry, in particular supersymmetry, to protect the self-energy contribution from growing too large—the approach we learned from the electron mass. Another one was inspired by the proton mass: the Higgs boson could be composed of lighter particles bound together by new strong interactions. Both strategies predict new particles around the energy scale of $10^3$ GeV and the search for them became one of the main goals of the Large Hadron Collider (LHC) at CERN. Herein lies the strength (and possible vulnerability) of naturalness. The new particles should be seen at this scale, or the Higgs mass becomes unnatural. We have an aesthetic principle that is being tested now. The non-observation of new physics at the first two LHC runs (2009–2018) has led on one side to a lively exploration of new alternatives to supersymmetry and compositeness, on the other side, it stimulated questions about the role of naturalness and other beauty criteria in particle physics.

Critics suggest the physics community became too focused on following the same aesthetic principles in theory building which led to the success of previous theories such as the Standard Model. According to critics, these principles may no longer apply. The universe can be slightly ugly and our emphasis on perfect symmetry may be prejudice rather than a physics idea [49]; it is not clear how one defines beauty [50]; our quest for top-down unification has significantly slowed down [51], and beauty may lead physics astray [11].

This concern and the clear traces of skepticism confirm that beauty plays an important role in fundamental physics. They also indicate that the naturalness problem is more mysterious than ever. The simplest solutions we tried are under serious tension. We could still see supersymmetry in a more involved form in Run 3 at the LHC, or there is a natural but quite unexpected solution for the Higgs mass—both options would strengthen the status of naturalness in our aesthetic canon. On the other hand, skeptics could be right and we should not put too much trust in naturalness. We could be in Kepler's shoes—the Higgs mass may not be a fundamental parameter and its value could be an accident without a natural explanation. This alternative is consistent, e.g., with multiverse theories that propose a huge number of universes, each with a different value of the Higgs mass, and we happen to live in one with $M_H = 125$ GeV$/c^2$. The multiverse approach seems to be in direct opposition to the platonists' view of the Universe that is knowable down to a unique prediction of all constants of nature. Moreover, it appears inconsistent with the scientific method—it is not clear how the multiverse hypothesis can be tested.

Whether nature continues its trend towards beauty as perfect unity, symmetry, and naturalness or takes a surprising, 'ugly' turn, remains to be seen. The LHC and future colliders will address this question. Their research program is therefore deep both on physical and philosophical grounds.

## 7. McAllister's Theory of Scientific Revolutions

In this section, we discuss the concept of dynamic beauty due to McAllister from a platonist position. McAllister establishes a possible connection from beauty to truth through aesthetic induction, leaving also an option to break the connection in a scientific revolution which consists of the radical change of aesthetic criteria [4].

McAllister argues that scientists use not only empirical but also aesthetic criteria to evaluate their theories. The aesthetic criterion is defined by an aesthetic property $P$ and its weighting $W_p$, the value of which represents how influential the property $P$ is in the evaluation of theories in a given time. The key point is that the value of $W_p$ can change up or down in response to the empirical success or failure of past and recent theories possessing $P$. In the normal period of science $W_p$ slowly increases in time if it takes positive feedback from the continued empirical success of the $P$-bearing theories (the mechanism called aesthetic induction). The collection of the most influential aesthetic criteria adopted by the community forms an aesthetic canon.

Naturally, when facing two new theories, both empirically successful, physicists prefer the one which conforms to the existing canon. This practice continues in the normal period of science up to a critical point when the non-conforming theories become empirically more successful than the conforming ones. This is the moment of the split in the community. The conservative supporters of the old canon oppose the non-conforming theories purely on the grounds of their ugliness. In case of continued empirical success of these unorthodox theories the opposition wanes, the weightings $W_p$, which shift slowly during the normal periods of science, drop quickly, and the old canon is eventually repudiated in favor of a new one, which is the essence of the scientific revolution. Beauty, which hinted at the truth in the normal period of science, has now led conservatives astray and hampered the progress of the field. McAllister mentions Kepler's divorce from the two-millennia-old canon of the circular planetary orbits and the abandonment of visualization and determinism in quantum mechanics by Bohr and Heisenberg as examples of scientific revolutions.

Important for McAllister's theory is the concept of beauty as a value. The statement that a theory or an object is beautiful involves a subjective evaluation by a scientist who projects aesthetic value to the theory based on how she views its importance, aptness, or future promise. This opens a possibility for the weightings $W_p$ to evolve in time—a concept of dynamic beauty allowing to explain revolutions as the change in the aesthetic canon.

Potential weaknesses of this influential theory were pointed out by several authors [3,5,6,32]. One of them is particularly relevant. Montano argues that certain aesthetic criteria, such as simplicity, behave as historical constants enjoying unwavering popularity despite the formulation of empirically successful theories which were far from

simple [6]. Why would they resist the change? While interpretations vary, we normally expect this kind of behavior in platonism.

Plato understands beauty not as a subjective value but as a property intrinsic to objects. Observing the same object/theory without subjective evaluation, scientists should agree on its beauty independently of time. We can best observe this intrinsic (objective) beauty when there is no advantage associated with it that we could in some way profit from, except for the pure joy of contemplating it. For example, I think that the gauge symmetry of the Standard Model will always have a proper amount of intrinsic attractiveness, even if its novelty, excitement it generates, promise, and expectations of further success (the subjective part which always sneaks in as science is done by humans) wear off. Likewise, General relativity strikes a chord with scientists regardless of its venerable age and insufficiency (not being a quantum theory). Arkani-Hamed says that the magnificence of the Newtonian classical physics peak does not evaporate when the quantum one is climbed. Objective beauty has an eternal quality.

This permanency of beauty applies to individual theories, however, things can appear different when we study a series of empirically successful theories adopted by the community over a certain period. Looking for a particular aesthetic property $P$, we might observe that objective beauty ($P$ and its weighting $W_p$) evolves from theory to theory and hence in time. In Section 5 I described how symmetry and unity gradually took center stage in particle physics. Starting as a mere ornament, symmetry became the cornerstone of our theories and unity has also gone a long way from the initial array of disconnected theories and phenomena. Respect for the two has undoubtedly grown and, by implication, so did their weighting in the canon. The growth of $W_p$, in this case, is not due to the subjective evaluation or habituation but due to the objective process: scientists uncover the full intrinsic beauty of nature in a step-by-step process, law by law. Plato's hierarchy of forms supports this picture. Physicists are gradually led through the discovery process from the world of senses to the knowledge of more and more beautiful versions of symmetry and unity at the higher levels of the world of forms.

At some point, we may get stuck. There is no guarantee that the aesthetic canon, manifested at the current level of the hierarchy and below, will succeed at the levels above. We liked visualization (and we still do) but we did not find it at the level of quantum mechanics and we do not expect it anymore to be a significant part of higher-level theories. It was replaced by symmetry and unity manifested at all currently known levels. Plato would project that they will succeed also above but we have to be prepared for all alternatives. If we hold on to our canon too tight, we may find ourselves in the shoes of the conservatives preventing the progress of the field, nearing the critical point of a revolution.

In an alternative new approach, which links beauty to understanding rather than truth (beauty as regulative ideal), Ivanova, drawing on examples by Hossenfelder [11] and adding some of her own, suggests that the case for the aesthetic induction is inconclusive [32], that we do not see a clear correlation between empirically successful theories and the change (positive or negative) in the acceptance of the corresponding aesthetic property found in these theories. As far as the majority of aesthetic properties is concerned, I agree there is not a clear trend of dynamic temporal development, the one I observe for unity and symmetry. Simplicity and naturalness appear constant which may indicate two different things. First, they are of different nature than symmetry and unity, being true constants; second, they do change up and down in response to individual theories but these changes are not resolved at the level of individual theories due to the subjective fluctuations among scientists mentioned above. The former option is consistent with the concept of the regulative ideal.

Most arguments by Hossenfelder are aimed at naturalness. She is right in pointing out that naturalness arguments related to the Higgs mass assume a uniform underlying distribution. The Higgs mass would not appear unnatural if the underlying distribution somehow favored the observed value of 125 GeV. We do not know the underlying distribution so our tacit assumption about it is—just an assumption of beauty. It hinges on the simplest (economic) choice of uniform distribution, linking naturalness to simplicity.

Regarding unity and symmetry, Hossenfelder acknowledges their exceptional successes, her objection is simply that "experience with last century's theories might not be of much help conceiving better ones" and that the better, empirically successful theories are slow to come. The first part of the objection is a useful reminder, the second part, however, is a subjective one. We waited almost 50 years to confirm the Higgs boson and 100 years to discover gravitational waves. The remaining difficult questions may take time to resolve. This should not be taken as an argument against beauty.

## 8. Conclusions

Fundamental physics appears to fit naturally into Plato's scheme in which beauty displays hierarchy in its forms. Physical objects and phenomena of the sensible world correspond to the lowest level, apparent beauty. The laws of physics represent higher forms, the true forms, real, objective, and unchanging. The objects and phenomena obey the laws and share some of their aesthetic features (some symmetry) but not all of them—the full symmetry of the laws is spontaneously broken. To see it, i.e., to reach the knowledge of it, requires effort whether from a physicist or, in the case of horse-breeding laws, from Weinberg's breeder.

Higher forms of beauty in this picture converge to the single final form, beauty as perfect unity, from which all other forms derive. The ever-increasing unification of the laws of physics indicates that we might be on a path to the final form in our field. Beauty is so important for Plato because it is clearly manifested in the sensible world and hence could serve as a means to the world of forms.

Throughout history, many leading physicists identified with Plato's views to various degrees. Beauty was an important value for them, either as motivation to do science, an important signpost to the truth, or a guide to understanding.

They used various aesthetic criteria along with empirical criteria to develop and judge theories. The current aesthetic canon has not been defined uniquely, however, symmetry, unity, and naturalness play a prominent role in it. In particular, symmetry is the pillar of our theories rather than a mere ornament. The use of the canon, along with empirical criteria, culminated in the formulation and experimental confirmation of the Standard Model of particle physics.

We learned that the laws of physics are elegant and this is not a trivial statement as Frank Wilczek pointed out. Some of my colleagues claim that whatever we find, it will be beautiful. I agree but not in the sense that physicists will automatically like anything as a reward for their effort. There is a clear difference between symmetry and asymmetry, unified and unrelated, natural and unnatural. Nature has been consistent in her choices so far—her laws are elegant although they did not have to be.

Now, with no beyond-the-Standard Model particles seen at the LHC, the plot thickens. All options are open for the naturalness problem. Conventional beauty, a new canon, and unattractive solutions such as fundamental asymmetry or multiverses. This adds to the already strong physics case for a future collider. Plato offers a picture of our world that is far more beautiful than we have seen so far. The question is simple. Is he right? Will beauty continue its status as the splendor of truth?

**Funding:** This research was funded by the Ministry of Education of the Slovak Republic via project FEPO.

**Data Availability Statement:** Not applicable.

**Acknowledgments:** I would like to thank the members of the International Particle Physics Outreach Group (IPPOG) for the discussions which inspired me to write this article.

**Conflicts of Interest:** The author declares no conflict of interest.

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
