# Peer review of "Aesthetic Criteria in Fundamental Physics—The Viewpoint of Plato"

_philosophies, doi:10.3390/philosophies7050096_

Round 1
Reviewer 1 Report
This is a very interesting article engaging with a very timely and interesting topic – the role of beauty in science. While there is a lot of interesting points, the paper can be significantly improved by engaging in more depth with prominent arguments in the literature and positioning itself a little better among already excising accounts.
First and most importantly, I worry how much this paper will convince anyone who has been following this debate over the last few years. To become more convincing, the article needs to engage more closely with philosophical arguments and ensure it situates itself in this literature by offering arguments as to why the existing positions in the debate are unsuccessful. For instance, James McAllister’s classic work is only mentioned briefly in the concluding remarks. This account needs to take more central place in the article and the author needs to offer an account as to whether they agree/disagree with the idea of beauty being a dynamic concept – something which would not align with a Platonist treatment that is offered in the article. How does one account for McAllister’s arguments? We need to see more analysis on this point if the article is to offer something original.
There needs to be more careful presentation of how scientists see the role of beauty. The author mentions (1) motivator and (2) guide to truth, but there is also a heuristic justification of beauty’s use which is not mentioned but needs to be (Ivanova (2017, 2020) is a good resource for these distinctions and relevant scientists who have defended such a position).
There are also some inaccuracies. The author argues that Poincare sees beauty as a motivator. This is not how Ivanova (2017) – referenced in the article – sees Poincare’s position. In the reconstruction she offers, Poincare seems to develop a much more sophisticated (Kantian) notion of the role of beauty as a condition of understanding. It would be good to pay more attention to this way of viewing the role of beauty and how a Platonist can square with it.
The latter position in the debate that is not mentioned in this article. But recently philosophers of science have developed exactly this thesis to diverse from the idea that beauty necessarily needs to be linked to truth to be epistemically valuable. Breitenbach (2017), Elgin (2020) and Ivanova (2020) have all argued we should see beauty as ‘gatekeeper’, a facilitator of understanding. It would be beneficial to engage with these arguments if this article is to offer am informed account of the role of beauty in science.
Last, the article should certainly not omit debating the arguments in Hossenfelder’s book that undermine the idea beauty can lead to truth, or at least undermines any inductive attempts to such justification. McAllister originally appeals to such inductive justifications. More recently Ivanova (2020) has argued we cannot use inductive arguments in this debate as they are inconclusive, so the role of beauty needs to be otherwise justified. To help advance this debate further, the article should engage more systematically with these justifications and offer a defence of why the Platonic model explored in the paper is still to be taken seriously.
Author Response
I would like to thank you for all the comments. I will address them point by point.
To become more convincing, the article needs to engage more closely with philosophical arguments and ensure it situates itself in this literature by offering arguments as to why the existing positions in the debate are unsuccessful. For instance, James McAllister’s classic work is only mentioned briefly in the concluding remarks.
Originally I had planned to include a section on McAllister, then decided to omit it. Now, prompted by you (a nice challenge), I thought it through and, as a result, have added a new section 7, McAllister's theory of scientific revolutions, to deal with this point. I do not focus too much on the claim that existing positions in the debate are unsuccessful, rather I offer an alternative. Beauty as a dynamic concept is at the centre of this section.
There needs to be more careful presentation of how scientists see the role of beauty. The author mentions (1) motivator and (2) guide to truth, but there is also a heuristic justification of beauty’s use which is not mentioned but needs to be (Ivanova (2017, 2020) is a good resource for these distinctions and relevant scientists who have defended such a position).
Thanks for pointing this out. I was not aware of Ivanova 2020 and this reference should not , indeed, be missing in my article. I have addressed the understanding of beauty as a heuristic tool and a pointer to understanding at several places: sec. 3 lines 153-154, 202-209, 223-225, sec. 7 lines 609-634.
There are also some inaccuracies. The author argues that Poincare sees beauty as a motivator. This is not how Ivanova (2017) – referenced in the article – sees Poincare’s position. In the reconstruction she offers, Poincare seems to develop a much more sophisticated (Kantian) notion of the role of beauty as a condition of understanding. It would be good to pay more attention to this way of viewing the role of beauty and how a Platonist can square with it. The latter position in the debate that is not mentioned in this article. But recently philosophers of science have developed exactly this thesis to diverse from the idea that beauty necessarily needs to be linked to truth to be epistemically valuable. Breitenbach (2017), Elgin (2020) and Ivanova (2020) have all argued we should see beauty as ‘gatekeeper’, a facilitator of understanding. It would be beneficial to engage with these arguments if this article is to offer am informed account of the role of beauty in science.
I fixed that Poincare innacuracy and for the role of beauty as a condition of understanding, see again sec. 3 lines 153-154, 202-209, 223-225, sec. 7 lines 609-634 mentioned above.
Last, the article should certainly not omit debating the arguments in Hossenfelder’s book that undermine the idea beauty can lead to truth, or at least undermines any inductive attempts to such justification. McAllister originally appeals to such inductive justifications. More recently Ivanova (2020) has argued we cannot use inductive arguments in this debate as they are inconclusive, so the role of beauty needs to be otherwise justified. To help advance this debate further, the article should engage more systematically with these justifications and offer a defence of why the Platonic model explored in the paper is still to be taken seriously.
I deal with this implicitly in the whole sec. 7 and explicitly in sec. 7, lines 622-634. Perhaps more space would be needed to discuss these points (and those above) in more detail but I felt it would get out of proportion with the rest of the text if I did it in this article.
Reviewer 2 Report
I was very happy to read this paper while reading it. It has extensive contents related to philosophy of physics which should be dealt with in learning physics. To be improved, I'd like to suggest several comments to the authors.
First, the title of the paper should be more elaborative. Even though this paper deals with aesthetics criteria of physics, the idea of the authors is rooted in Plato's thoughts. In fact, there have been a lot of views about the aesthetics, e.g., Plotinus, Shaftesbury, Hutcheson and Kant. This paper does not deal with the whole views on aesthetics. In this light, I think the authors should articulate their theoretical lens adopted in this paper. Like, "Aesthetic consideration of fundamental physics from the viewpoint of Plato".
Second, I hope the authors will give readers more explanation about the contents. For example, the paper mainly discusses unity, symmetry and naturalness. Nevertheless, there are a number of criteria related to aesthetics (elegance, sublimity, harmony, etc). The authors should explain why those are ruled out in this paper. As well, the criteria mentioned in this study have somewhat fallacies. For example, symmetry is a key concept in quantum mechanics however it is occasionally wrong: e.g., symmetry breaking in QED. Even, it is really difficult to define the term of symmetry and to somewhat subjective. The authors should clarify the limitations of this study and you may leave them for the further discussion.
Third, I suggest to narrow down the specific field of physics. The author stated 'fundamental physics' but I think the main field is quantum mechanics. It would be helpful for readers to understand the intention of the authors.
Author Response
Thank you for all your helpful suggestions, I address them one by one:
1. First, the title of the paper should be more elaborative. Even though this paper deals with aesthetics criteria of physics, the idea of the authors is rooted in Plato's thoughts. In fact, there have been a lot of views about the aesthetics, e.g., Plotinus, Shaftesbury, Hutcheson and Kant. This paper does not deal with the whole views on aesthetics. In this light, I think the authors should articulate their theoretical lens adopted in this paper. Like, "Aesthetic consideration of fundamental physics from the viewpoint of Plato".
Agreed. I changed the title to "Aesthetic criteria in fundamental physics - the viewpoint of Plato".
2. Second, I hope the authors will give readers more explanation about the contents. For example, the paper mainly discusses unity, symmetry and naturalness. Nevertheless, there are a number of criteria related to aesthetics (elegance, sublimity, harmony, etc). The authors should explain why those are ruled out in this paper. As well, the criteria mentioned in this study have somewhat fallacies. For example, symmetry is a key concept in quantum mechanics however it is occasionally wrong: e.g., symmetry breaking in QED. Even, it is really difficult to define the term of symmetry and to somewhat subjective. The authors should clarify the limitations of this study and you may leave them for the further discussion.
I added a text in sec. 4, lines 285 - 294, to motivate my choice of criteria. Basically, it is difficult to discuss extrinsic criteria such as elegance, sublimity, and wonder due to their subjective nature. Unity and symmetry are, as I see it, objective, well-defined and they are clearly manifested and the most influential in our theories. As to symmetry being subjective (your claim), I am not sure, perhaps outside what I call fundamental physics. As I see it, Group theory defines symmetries we use uniquely. Symmetry breaking could be explicit or spontaneous (I am not sure which particular symmetry and symmetry breaking in QED you mean) and the latter one provides a beautiful connection between symmetric laws and the symmetry breaking solutions of the laws, as disscussed in the article.
Third, I suggest to narrow down the specific field of physics. The author stated 'fundamental physics' but I think the main field is quantum mechanics. It would be helpful for readers to understand the intention of the authors.
Fair enough. I added an explanation in sec. 1, line 52-54: "By fundamental physics I mean the physics of the four fundamental interactions: electromagnetic, strong, weak, and gravitational, i.e., particle physics plus gravity."
Round 2
Reviewer 1 Report
The author has addressed my comments and the manuscript is much improved, suitable to be published in it s current form.